# Preliminary Analysis of a New Power Train Concept for a City Hybrid Vehicle

**Roberto Capata** 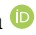

Department of Mechanical and Aerospace Engineering, University of Roma "Sapienza", 00185 Roma, Italy; roberto.capata@uniroma1.it; Tel.: +39-06-44585243

**Abstract:** This research aims to test the feasibility of a prototype of a newly designed thermal engine for a hybrid propulsion vehicle. This study consists of the implementation of an innovative supercharger for city car internal combustion engine ICE (900 cc). The preliminary proposal presented here is to mechanically disconnect the compressor/turbine device, supporting the rotation of the compressor with a dedicated electric motor and connecting a turbine to a generator. Mechanical decoupling will allow both machines to be designed for operating closer to their maximum performance point, for most of the expected real field of operation. Specifically, the turbine is likely to have a lower rotation speed than the original group and will, therefore, be slightly larger. The advantage is that, while in the current supercharger groups the surplus at high regimes is discharged through the waste-gate valve without expanding in a turbine, in the configuration proposed, all the energy of the combustible gases is used by the turbine to generate electrical power that can be used where required. Once the motorization of the vehicle (999 cc) has been fixed, the two turbomachines will have to be studied and designed, looking where possible, for commercial components. Finally, a computational fluid dynamic CFD will be needed to verify the validity of the choice, followed by careful experimentation campaigns.

**Keywords:** mild-hybrid; power train; compressor; turbine; operative maps

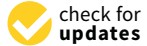



## 1. Introduction

Turbocharging is the practice of compressing the air before injecting it into the cylinder (thus increasing its density, especially if an intercooler is also used), and of exploiting the energy of the exhaust gases to recover enough power in a gas turbine to drive the compressor. This practice is vastly adopted in diesel engines: in low-speed, heavy-duty vehicles, it helps produce the required high powers and torques, while in fast running, passenger cars, it boosts the available power, making the vehicle more performant. In smaller gasoline engines, its main function is to boost the specific power (kW/cc). However, the turbocharger does not deliver the necessary boost pressure until the (usually centrifugal) compressor reaches speeds of 60,000–100,000 rpm, thus allowing full engine power to be produced. Typically, the turbocharger can take 3 to 5 s to run-up to speed, so there is significant time between the driver demanding full engine power and it being available (turbo lag). The term "turbo-compounding" (TC in the following) also refers to the use of a turbine to recover energy from the gas exhaust system of an internal combustion engine (ICE) and reintroduce it back into the engine, usually as mechanical energy (overpressure at the inlet) but includes different alternative ways for doing so.

There are currently two forms of TC: mechanical, introduced in the aeronautical sector as early in the 1940s, and electrical, a newer arrangement that was made possible by the advance of mechatronics and whose development is encouraged by the ever-higher electrical power absorption in modern vehicles. In mechanical TC, the waste gas potential (total enthalpy) energy is partially recovered in a turbine located downstream of the engine exhaust, and may be fed back into the engine in two ways: directly in the transmission line

via a high reduction gearing, or indirectly, by driving an air compressor that injects air at a higher pressure into the cylinder (as in "normal" turbocharging). In electrical TC, the recovered energy is distributed using a power electronic module to the on-board systems (including possibly the supercharger). The mechanical TC technology has been utilized both in diesel and gasoline (and gas) engines, despite the inherent complexity of the system hardware and its relative "fragility" (a common failure being the turbine breakup because of extreme thermal cycling). However, the increasing demands of ancillary electrical equipment and the rise of hybrid powertrain vehicles have led to a growing electrical energy requirement in the vehicle, and as a result, since the electric turbo-compounding system offers a better solution, it is attracting more attention.

The success of either type of TC depends on a suitable design of the gas turbine unit, which must have a low back pressure over its entire operating range, good efficiency, and the necessary sturdiness to withstand the continuous variations of the mass flow rate of the exhaust gases and their temperature. It is generally acknowledged that the electrical TC offers significant advantages over its mechanical counterpart, but its intrinsic complexity and the need for an accurate, reliable, and properly tuned electronic controller makes its commercial use still problematic. The idea developed in this project envisions the use of an electrical TC [1,2], with the turbine driving an electrical generator to produce electrical energy for charging the battery pack and possibly feeding the onboard utilities (such as oil and water pumps, cooling fans, etc.). In a hybrid-propulsion vehicle, energy storage (battery pack) is also incorporated into the system to supply the high power pulse required by this machine. With such an arrangement, the vehicle can also incorporate a kinetic energy recovery system (KERS) to recover vehicle braking energy that can also be stored in the energy store for use during the next acceleration phase. The actual rating of the electrical machine, energy store, and associated power electronics, and will depend on the vehicle operating cycle.

## 2. The Proposed Task

The recent technical literature reports overall engine efficiency improvements of TC-equipped ICE in the range of 3–5%. Besides, both theoretical and simulation studies have shown that engine performance improves only at a slower speed. Poor performance at higher engine speeds is recorded due to excessive exhaust back pressure by the power turbine [3,4].

Therefore, a significant amount of resources is being invested in an attempt to improve the existing turbines characteristic curve.

Synthetically, the current state of the art is the following:

(1) For turbine stages, the focus is on the investigation and development of technologies that would improve on-engine exhaust energy utilization compared to the conventional radial turbines in widespread use today;

(2) For compressor stages, the focus is on investigating compressor design parameters beyond the range typically utilized in production (i.e., higher pressure ratios), to determine the potential efficiency benefits thereof;

(3) For TC units, the focus is on the development of a robust bearing system that would provide higher bearing efficiencies compared to systems used in turbo-compound power turbines in production.

The proposed task is to address the above problems and find a suitable solution that can be immediately implemented on a real gasoline ICE.

## 3. Turbocompressor Specifications

Once the vehicle, and consequently the engine (999 cc) has been established, the existing group on the vehicle has been identified: the compressor currently installed is the Garrett@ gas turbine (GT) 12 model (the operational map of the compressor is shown in Figure 1). The procedure to define the compressor is as follows [5–8]:

1.　It calculates the design volumetric suction flow rate (calculated at "standard conditions" specified by design standards);
2.　Once the flow rate and the pressure ratio are known, the preliminary "shape" of the compressor occurs; the specific work is given by:

$$L = T_0 \left( c_{p,2}\, \beta^{\frac{k-1}{k \cdot \eta_p}} - c_{p,o} \right) \tag{1}$$

3.　Based on the type of compressor, the tentative maximum peripheral speed, $U_{max}$, is fixed and the necessary Euler's work is obtained:

$$W_{EUL} = U_{max} V_{2t} = U_{max}^2 \psi_2 \tag{2}$$

4.　If the value of the $\psi_2$ obtained from the calculation is acceptable (i.e., falls within the field of values historically adopted in similar machines with technically satisfactory results), only one stage will be sufficient. Otherwise, there are two possibilities:

(a)　To increase the $U_{max}$, choosing a different geometry and return to step 3;
(b)　To keep the specification values for $U_{max}$ and $\psi_2$, and calculate the number of stages from the equation:

$$N_{stages} = \frac{W_s}{W_{EUL}} \tag{3}$$

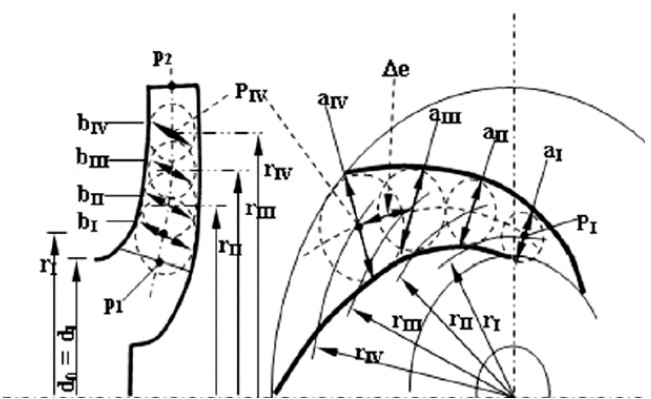

**Figure 1.** Rotor channel tracking.

Below are the steps and formulae used to design the compressor:
**Fluid dynamic equations**

$$\text{Inlet flow rate}: \ Q = \pi\, r_{1e}^2 \left( 1 - \chi_1^2 \right) U_1\, \varphi_1\, \delta_{1p} \tag{4}$$

$$\text{Outlet flow rate}: \ Q = 2\, \pi\, b_2\, r_2\, U_2\, \varphi_2\, \delta_{p2} \tag{5}$$

$$\text{Euler equation}: \ L_{EUL} = gH = U_2 V_{2t} - U_1 V_{1t} \tag{6}$$

**Auxiliary equations**

$$\text{Inlet peripheral velocity}: \ U_1 = \omega\, r_1 \tag{7}$$

$$\text{Outlet Peripheral velocity}: \ U_2 = \omega\, r_2 \tag{8}$$

**Thermodynamic equations**

$$\text{Fluid state equation}: \ \frac{p}{\rho} = ZRT \tag{9}$$

$$\text{Isentropic transformation :} \quad T_2 = T_1 \beta^{\frac{k-1}{k}} \tag{10}$$

**Equations imposed by constraints**

$$\text{structural limitation :} \quad U_2 < U_{max} \tag{11}$$

$$\text{limitation due to transonic phenomena :} \quad W_1 = 0.8 \div 0.85 c_s \tag{12}$$

$$\text{the suction pre} - \text{rotaion law :} \quad V_{1t} = 0 \tag{13}$$

$$\text{a relationship between } \psi, \varphi, r_1 \text{ and } r_2$$

Thus there are 19 unknowns − 12 equations − (Q, $L_s$) = five degrees of freedom. The designer must freely choose five of the kinematic and geometric quantities that identify a turbocharger. In this case, the following values are fixed: $\psi_2$ =1 (rotor radial outlet), $\varphi_2$ = 0.55 (actually the range for radial compressor is between 0.3 and 0.6), $\varphi_1$ = 0.3 (normal range is within 0.1 and 0.3), and axial inlet is adopted ($\psi_1$ = 0) and, finally, a limitation on $U_2$ is set (<200 m/s).

In radial turbochargers, the blades are double curvature, although there are numerous examples with simple curvature or—for very small rated powers—even with no curvature (straight blades). The degree of reaction is given by:

$$R_\rho = \frac{L_{EUL} - \frac{V_2^2 - V_1^2}{2}}{L_{EUL}} = 1 - \frac{\psi_2^2 - \varepsilon^2 \psi_1^2}{(\psi_2 - \varepsilon^2 \psi_1)} - \frac{\varphi_2^2 - \varepsilon^2 \varphi_1^2}{(\psi_2 - \varepsilon^2 \psi_1)} \tag{14}$$

This relationship, as we can see, links the degree of reaction to $\psi$ and $\varphi$.

With regard to the definition of the shape of the rotor channel:

(a) the channel projection on the meridian section is calculated using a midline drawn by polynomial interpolation methods. The exact shape of the tracks on the meridian plane of the intersections of the hub and the case is obtained in two stages. In addition to any variations of $\delta_p$ with s, that can be (in the first approximation) neglected, the volumetric flow Q varies with s, due to the variation in the specific volume of the fluid during compression. The trend of p with s cannot be calculated exactly, and therefore it is estimated from the transformation diagram in the s/h plane. Once the trend of the passage area along the meridian line is known, the envelope of the circles of radius b(s) = A(s)/(2πr) is constructed. These two envelopes (one on the case side and the other on the hub), make up the walls of the canal.

(b) The shape of the channel in the front view is based on the definition of a midline too: both blades with an Archimedes spiral (logarithmic) as the front profile can be adopted, that is, with the characteristic of forming a constant angle θ with the radius throughout its development, as well as profiles of arc circle blades or polycmetric arcs. Having chosen the number of blades, the front section remains automatically defined.

The number of blades is chosen on the basis of empirical formulae obtained as places of maximum performance for certain geometric and kinematic characteristics:

$$z_p = \frac{\beta_2}{3} \quad (Stepanov) \tag{15}$$

$$z_p = K \cdot \frac{\sin\left(\frac{\beta_2 + \beta_1}{2}\right)}{\ln\left(\frac{r_2}{r_1}\right)} \quad with \ K = 14 \div 18 \quad (Eckert \ and \ Schnell) \tag{16}$$

Based on the design procedure now described, we obtain the geometric characteristics of the compressor shown in Figures 1 and 2 and Table 1 [9–12]. Of course, all speeds are set based on the measured values on the GT12 compressor. The operating points have also been fixed and drawn based on the maps of the commercial model. The operation specifications are as follows:

$\beta$ = 1.5

$\dot{m}$ = 0.02–0.06 kg/s

n = 140,000–210,000 rpm (corresponding to ICE minimum and maximum rotational speed assumed equal to 2000–5000 rpm, respectively).

From the data analysis [13], the correspondence to the commercial model can be noticed and already installed on the car (in this case the GT12 model). It then assessed the compressor operational at various ICE engine regimes using the map (Figure 3). Therefore, it is possible to compile the following table and represents the required power in function of ICE RPMs (Table 2 and Figure 4).

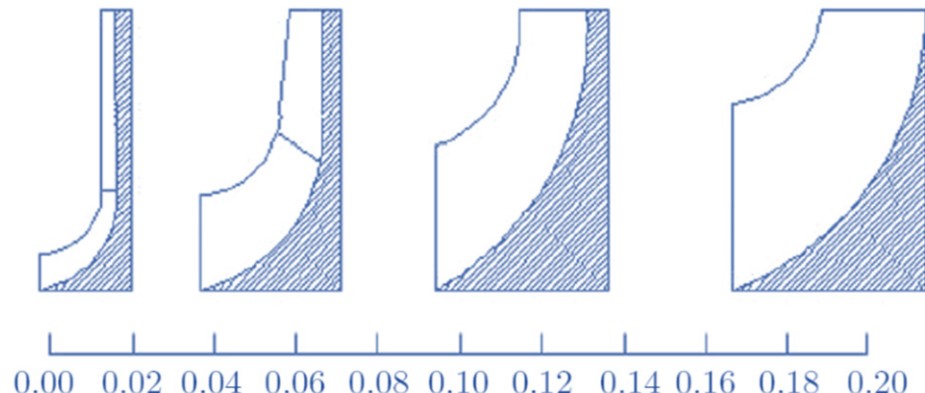

**Figure 2.** Typical channel shapes as a function of flow coefficient.

**Table 1.** Designed compressor specifications.

| $\dot{m}$ [kg/s] | 0.02 | $W_{EUL}$ [J/kg] | 44,652.74 | $\psi_2$ | 1 |
|---|---|---|---|---|---|
| $\beta$ | 1.4 | $U_2$ [m/s] | 211.31 | $\psi_1$ | 0 |
| $c_p$ [J/kg K] | 1004 | $r_2$ [m] | 0.0144 | $\varphi_1$ | 0.3 |
| $T_1$ [K] | 293 | $\Delta T$ | 22.2 | $\varphi_2$ | 0.55 |
| $\omega$ [rad/s] | 14,653 | $\rho_1$ [kg/m³] | 1.20 | $\delta_p$ | 0.98 |
| $\varepsilon$ | 0.42 | $\dot{Q}_1$ [m³/s] | 0.0167 | $R_\rho$ | 0.5 |
| $p_1$ [Pa] | 101,000 | $r_{1e}$ [m] | 0.0110 | $\chi$ | 0.65 |
| $T_2$ [K] | 337 | $r_{1i}$ [m] | 0.0072 | $(1 - \chi^2)$ | 0.58 |

**Table 2.** Compressor (GT)12 compressor operation at different internal combustion engine (ICE) engine rotation regimes.

| rpm ICE | $\dot{m}$ [kg/s] | rpm | $\beta_c$ | $T_{in}$ [K] | $T_{out}$ [K] | P [W] | $\eta$ |
|---|---|---|---|---|---|---|---|
| 2000 | 0.021 | 145,000 | 1.42 | 298.4 | 344 | 920.71 | 0.68 |
| 3500 | 0.0408 | 180,000 | 1.64 | 297.5 | 354 | 2474.39 | 0.8 |
| 5500 | 0.0619 | 210,000 | 1.84 | 297.5 | 366.8 | 4444.96 | 0.82 |

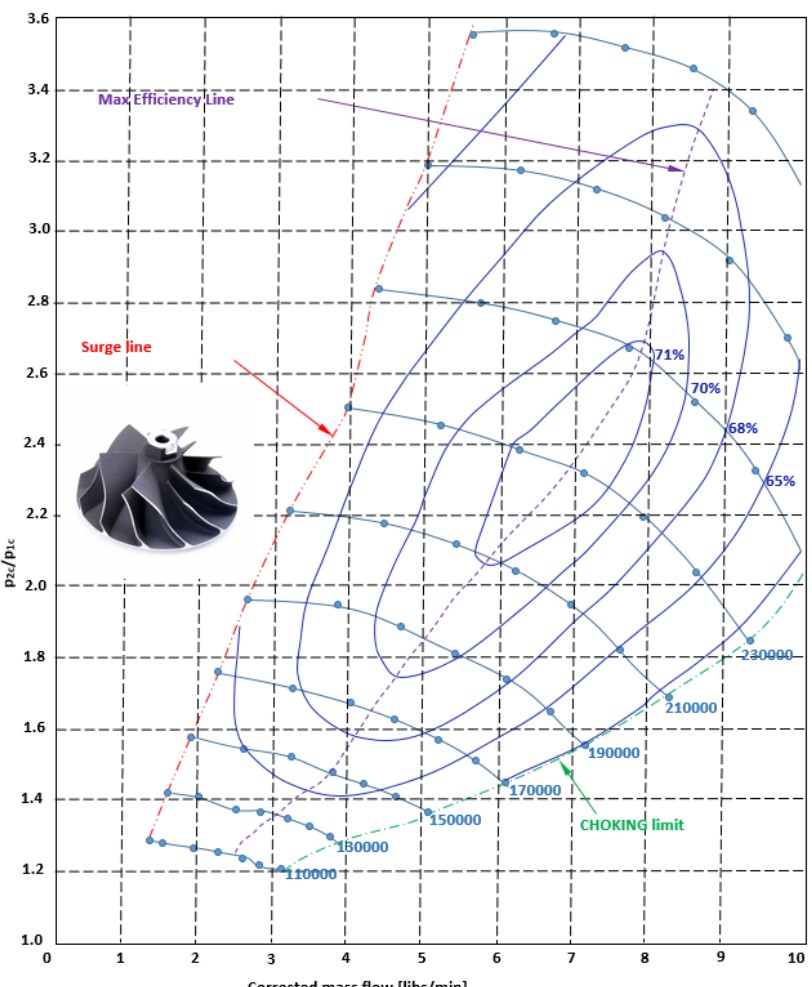

**Figure 3.** Operative compressor map.

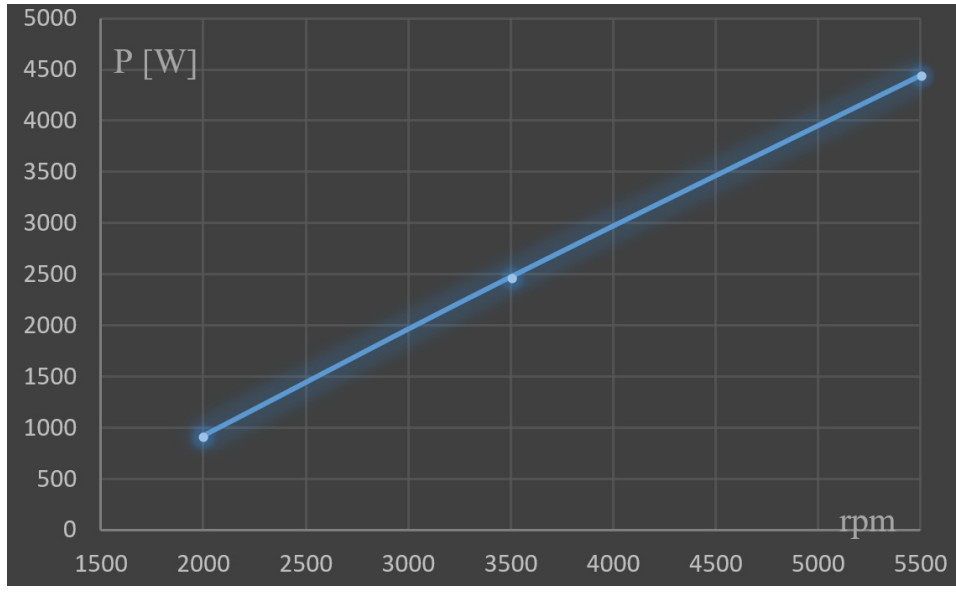

**Figure 4.** Required power trend depending on the ICE engine rotational speed.

### 4. Turbine Definition

Pictured is a typical layout of a radial gas turbine with radial input; the (absolute) construction angle of the shovel at the entrance is generally zero, a fact dictated by the resistance of the material and the high temperature of the gases. Rotor blades are subject to high levels of stress caused by centrifugal forces, along with stresses due to pulsating and therefore inherently non-stationary high-temperature gas. From Figure 5, the rotor palettes radially extend inwards and rotate the flow in an axial direction, decreasing its absolute tangential speed. The outer zone of the outgoing paddle is named the exducer and is curved to remove most if not all of the tangential component of the absolute speed. The radial turbine or centripetal turbine is very similar in appearance to the centrifugal compressor, but with the direction of flow and movement of the opposite blades.

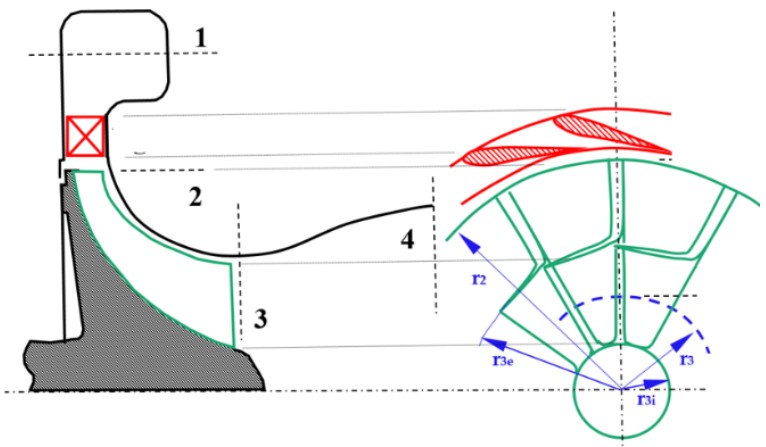

**Figure 5.** Inlet forward radial turbine reference layout.

Acting in the same way, a common $n_s/d_s$ procedure is adopted [14–17]. Clearly, the preliminary hypotheses are as follows:

- the motion is permanent and stationary;
- the speed is constant on the exhaust section, or, if post-rotation is adopted, the law of variation of $V_{2t}$ with the radius is known;
- the motion in the fixed channels takes place by cylindrical flaps, and in the mobile channels for helical flaps;
- the speed is constant on the supply section.

The equations available are, here too:
**fluid dynamic equations:**

$$\text{inlet flow rate}: \; Q_2 = \frac{m}{\rho \cdot (p_1 \cdot T_1)} = 2 \cdot \pi \cdot r_2 \cdot b_2 \cdot U_2 \cdot \varphi_2 \cdot \delta_{p2} \tag{17}$$

$$\text{outlet flow rate}: \; Q_3 = \frac{m}{\rho \cdot (p_2 \cdot T_2)} = \pi \cdot r_3^2 \cdot \left(1 - \chi_3^2\right) \cdot U_3 \cdot \varphi_3 \cdot \delta_{p3} \tag{18}$$

$$\text{Euler's work}: \; L_{EUL} = U_2 V_{2t} - U_3 V_{3t} \tag{19}$$

**auxiliary equations:**

$$\text{inlet peripheral speed}: U_2 = \omega r_2 \tag{20}$$

$$\text{outlet peripheral speed}: U_3 = \omega r_3 \tag{21}$$

**thermodynamic equations:**

$$\text{fluid state equation}: p/\rho = ZRT \tag{22}$$

$$\text{equation that relate } p \text{ and } T : \quad \frac{p_3}{p_2} = \left(\frac{T_3}{T_2}\right)^{\frac{\kappa}{\kappa-1}} \tag{23}$$

**equations imposed by constraints:**
limiting the exhaust speed

$$\text{to avoid transonic phenomena: } W_2 = 0.9 \, c_s \tag{24}$$

$$\text{law of rotation at the discharge: } V_{3t} = 0 \text{ or } V_{3t} = f(r) \tag{25}$$

$$\text{peripheral velocity limit: } U \leq U_{max} \tag{26}$$

Therefore, in the absence of other design constraints and in the hypothesis of knowing "a priori", the values to be given to $\delta_{p3}$ and $\delta_{p2}$, the unknowns are $r_2$, $r_{3i}$, $r_{3e}$, $b_2$, $U_2$, $U_3$, $V_{2m}$, $V_{2t}$, $V_{3m}$, $\omega$, $\dot{m}$, Ma, $p_2$, $p_3$, $T_2$, $T_3$, $r_2$, $r_3$.

19 unknowns $-$ 10 equations $-$ (m, $p_0$, $T_0$, $p_2$) = five degrees of design freedom

That is, the designer must choose the values of five between the geometric, kinematic and termodinamic quantities that identify the inlet forward radial turbine (IFR). The most convenient choice is to impose a $Ma_2 \ll 1$ to have an acceptable $Ma_3$, the $U_{max}$ value will be adopted for $U_2$, a precise law of variation of the Vt with the radius will be imposed (usually, the free vortex, $Vt\cdot r =$ constant); then, first-attempt values (derived from experience and comparison with existing models) are chosen. The design procedure is an iterative and sometimes it is needed to change or assign a new value to the chosen parameters. In line with the criteria for applying similarity to turbomachinery, even with regard to these choices, the design experience has led to the codification of "optimal" values (i.e., more convenient) of some of these quantities according to the shape of the current: here too, however, and for the same reasons, the available data are scarce. A series of recent experiments has led to the drafting of curves, named Chen and Baines, shown in Figure 6. It can see how yields are maximum for:

$$\psi_2 = 0.9 \div 1 \tag{27}$$

$$\varphi_c = \varphi_3 \, r_3 / r_2 \approx 0.2 \div 0.3 \tag{28}$$

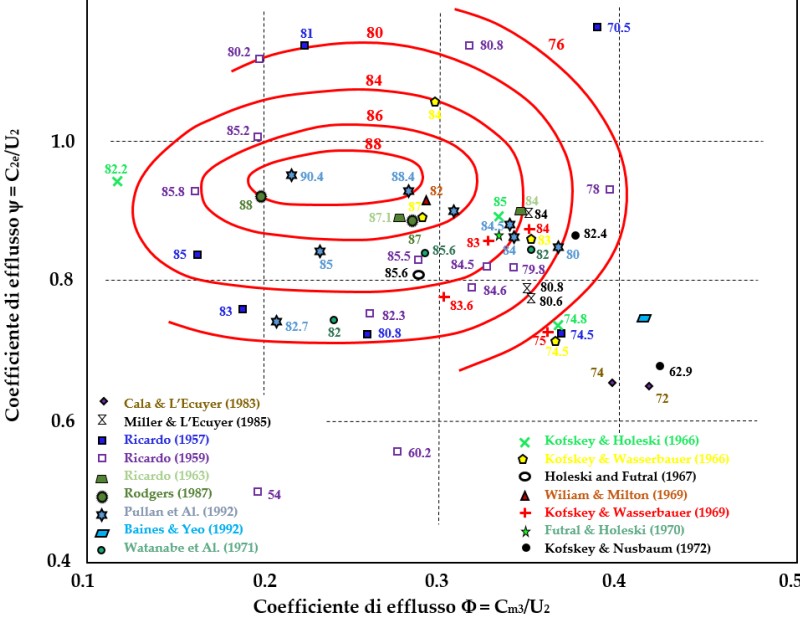

**Figure 6.** Chen and Baines maps.

There have also been curves from Baines (Figure 7) which show the efficiency as a function of the U/V$_s$ ratio, where $V_s = (2\Delta h)^{1/2}$ is the stagnation velocity equivalent to the entire enthalpic leap processed in turbine. $V_s$ is a function of the expansion ratio

$$V_s = \sqrt{2\Delta h} = \sqrt{2\cdot\left(c_{p1}T_1 - c_{p3}T_3\right)} = \sqrt{2c_{p1}T_1\cdot\left(1 - \frac{c_{p3}}{c_{p1}}\cdot\beta^{\frac{1-\kappa}{\kappa}\cdot\eta_s}\right)} \tag{29}$$

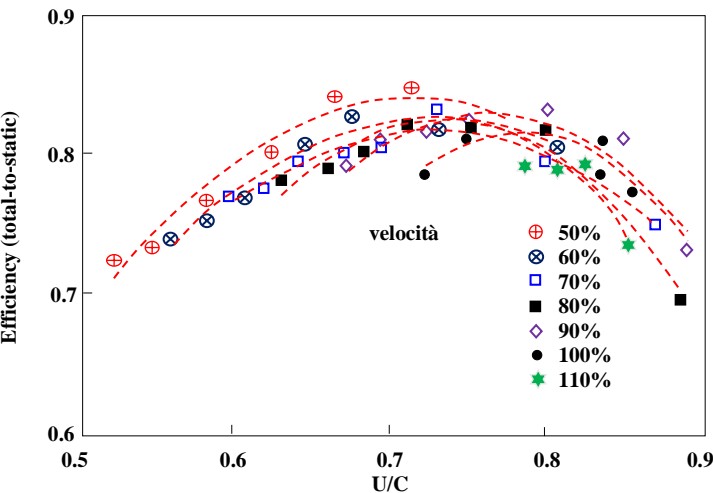

**Figure 7.** Baines chart.

The graphs in Figure 8 show that the various experimental curves all have a mass for U/V$_s$ = 0.7, and therefore this can become a reliable design criterion. In fact, these general choices are reflected in the operating curve of the Garrett GT20, confirming the general validity of the procedure used here. Radial IFR blades are usually dual curvature surfaces; the general expression for the degree of reaction is:

$$R_\rho = 1 - \frac{1}{2}\psi_2 \tag{30}$$

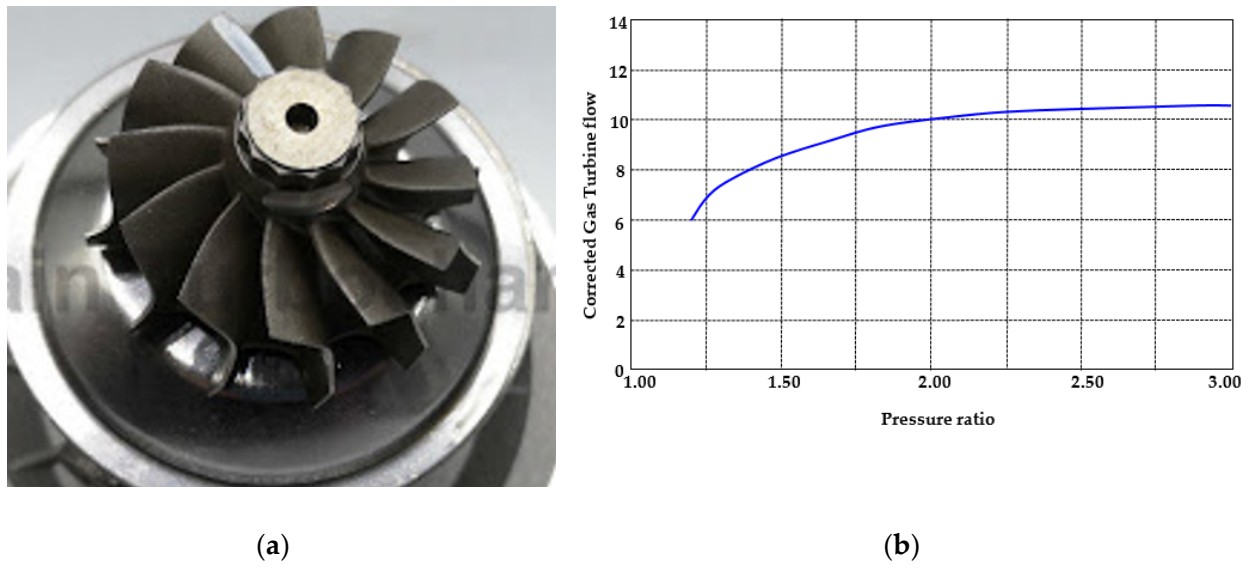

(**a**)          (**b**)

**Figure 8.** (**a**) GT 20 impeller, (**b**) operative map.

Now, for reasons of structural strength, in radial turbines the blades always have radial direction at the turning inlet, so you will have $\psi_2 = 1$ and then $R_\rho = 0.5$. With regard to the shape of the rotor channel, the following considerations can be made:

(a) the general shape of the channel in the meridian section is calculated on the basis of an average meridian line drawn by appropriate polynomial interpolation methods in a manner similar to what is seen in turbocompressors;

(b) the exact shape of the traces on the meridian plane of the intersections of the hub and the case is obtained in two stages: first the law of the passage area is fixed, it is established as the area useful for the disposal of the various flow rate with the curvilinear axle measured along the medium meridian line. In fact, it is advisable to fix the law of variation of the meridian velocity Vm. Fixing A(s) the area at the abscissa s, results A(s) = Q/(Vm δp). In this case, the volumetric flow Q varies with s, due to the variation in the specific volume of the fluid during expansion. Finally, the envelope of the radius circles b(s) = A(s)/(2nr) is constructed: these two envelopes (one on the case side and the other on the hub side), make up the walls of the channel. The shape of the channel, in the front view, is also based on the definition of a midline: the profiles adopted are all chosen on the basis of structural resistance considerations, and therefore have the least possible curvatures. Widely used as a front profile is the Archimedes spiral (logarithmic), but mixed profile blades are also used, with a straight and a circular arch stroke. Having chosen the number of blades, the front section remains automatically defined.

The blades number $z_p$ is chosen based on the consideration (Jamieson) that consider the ideal (non-viscous) flow in a rotor channel. The formula is:

$$z_p \;=\; 2\pi\omega r/V_{2m} \;=\; 2\pi U/V_{2m} \;=\; 2\pi V_{2t}/V_{2m} \;=\; 2\pi/\tan\alpha_2 \tag{31}$$

The criterion is quite restrictive and therefore the number of real blades is usually determined by:

$$z_{p,real} \;=\; z_p - (2 \div 3) \tag{32}$$

with the data in our possession the following values are set: $\psi_1 = 1$(rotor radial inlet), $\varphi_2 = 0.4$ (actually the range for radial compressor is between 0.4 and 0.55) $\varphi_1 = 0.3$ (normal range is within 0.1 and 0.3), and axial outlet is adopted ($\psi_2 = 0$) and, finally, the turbine efficiency is set from the available map ($h = 0.82$). The design results are shown in Table 3, is as follows.

**Table 3.** Turbine specifications.

| $\dot{m}$ [kg/s] | 0.0021 | $W_{EUL}$ [J/kg] | 103,194.4 | $\psi_1$ | 1 |
|---|---|---|---|---|---|
| $\beta$ | 1.4 | $U_2$ [m/s] | 321.24 | $\psi_2$ | 0 |
| $c_p$ [J/kg K] | 1414 | $r_2$ [m] | 0.051 | $\varphi_1$ | 0.3 |
| $T_1$ [K] | 980 | $\Delta T$ | 36.4 | $\varphi_2$ | 0.4 |
| $\omega$ [rad/s] | 6280 | $\rho_2$ [kg/m$^3$] | 0.39 | $\delta_p$ | 0.98 |
| $\varepsilon$ | 0.23 | $\dot{Q}_1$ [m$^3$/s] | 0.0054 | $R_\rho$ | 0.5 |
| $p_2$ [Pa] | 101,000 | $r_{2e}$ [m] | 0.0112 | $\chi$ | 0.65 |
| $T_2$ [K] | 907 | $r_{2i}$ [m] | 0.0072 | $(1 - \chi^2)$ | 0.58 |

From the data analysis, it can enlighten the correspondence to the GT20 model (Figure 8a shows the model and Figure 8b the operative map) [18]. It then assessed the operation of the turbine at the various ICE engine regimes using the available turbine map. The results are shown in Table 4.

**Table 4.** GT20 turbine operation at different ICE engine rotation regime.

| rpm ICE | $\dot{m}$ [kg/s] | rpm | $\beta_e$ | $T_{in}$ [K] | $T_{out}$ [K] | P [W] | $\eta$ |
|---|---|---|---|---|---|---|---|
| 2000 | 0.024 | 82,170 | 1.2 | 954 | 922 | 780 | 0.87 |
| 3500 | 0.04 | 134,483 | 1.6 | 1005 | 930 | 3510 | 0.86 |
| 5500 | 0.07 | 164,002 | 2.00 | 1045 | 932 | 9500 | 0.81 |

By acting in the same way as described above, we obtained the following performance of the power delivered by the turbine according to the rotational speeds of the ICE (Figure 9).

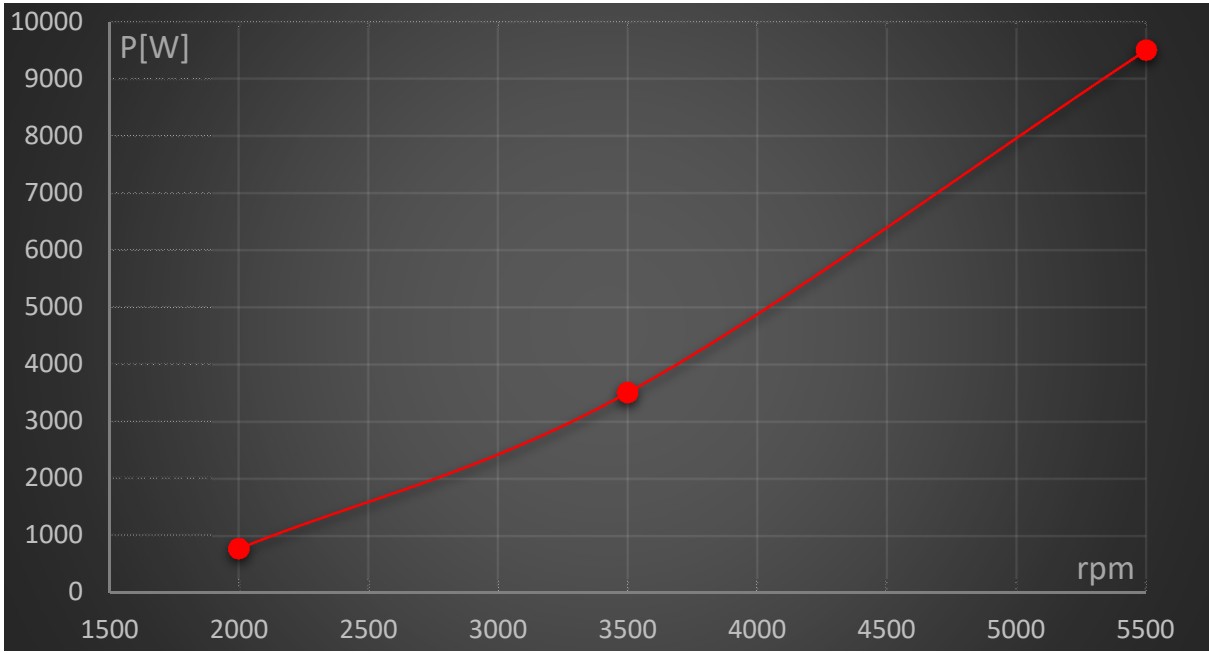

**Figure 9.** Power performance as a function of the speed of rotation of the ICE engine.

## 5. Preliminary Analysis and Future Development

Once the two maps were obtained, a match was performed between the two curves. Figure 10 shows this overlap. It can be noticed that, only in the first and short stretches, the turbine does not have enough energy to power the compressor. Starting with 2200 rpm, the turbine can generate power to move the compressor. Additionally, after the 3500 rpm, the difference is greater than 1 kW. This energy surplus is either able to recharge the battery pack of the vehicle or feed any onboard auxiliaries (GPS, check panel, etc.). These preliminary calculations confirm the goodness of the choice to decouple the turbocharger group. Besides, for the short period in which the turbine is not able to provide sufficient power to the compressor, the latter can draw energy from the battery pack, which will need to be carefully studied and designed [19,20].

The next step will be to simulate both a city and an extra-urban driving cycle, to see the trend of net power available to the turbocharger and the gain that this new configuration will make to the vehicle, in terms of fuel economy and therefore emissions [21–23].

At the same time, 3D models of the compressor and turbine will be constructed with a suitable solid modeler (SOLIDWORKS or the like) and computational fluid dynamic (CFD) simulations will be run to study the performance of both compressor and turbine under different operational loads.

Finally, a preliminary system layout will be devised and numerically simulated, using current technology performance parameters. The result will be the identification of a new

set of operating parameters (pressure ratio, rotational speeds, exhaust pressure) that lead to performance improvement.

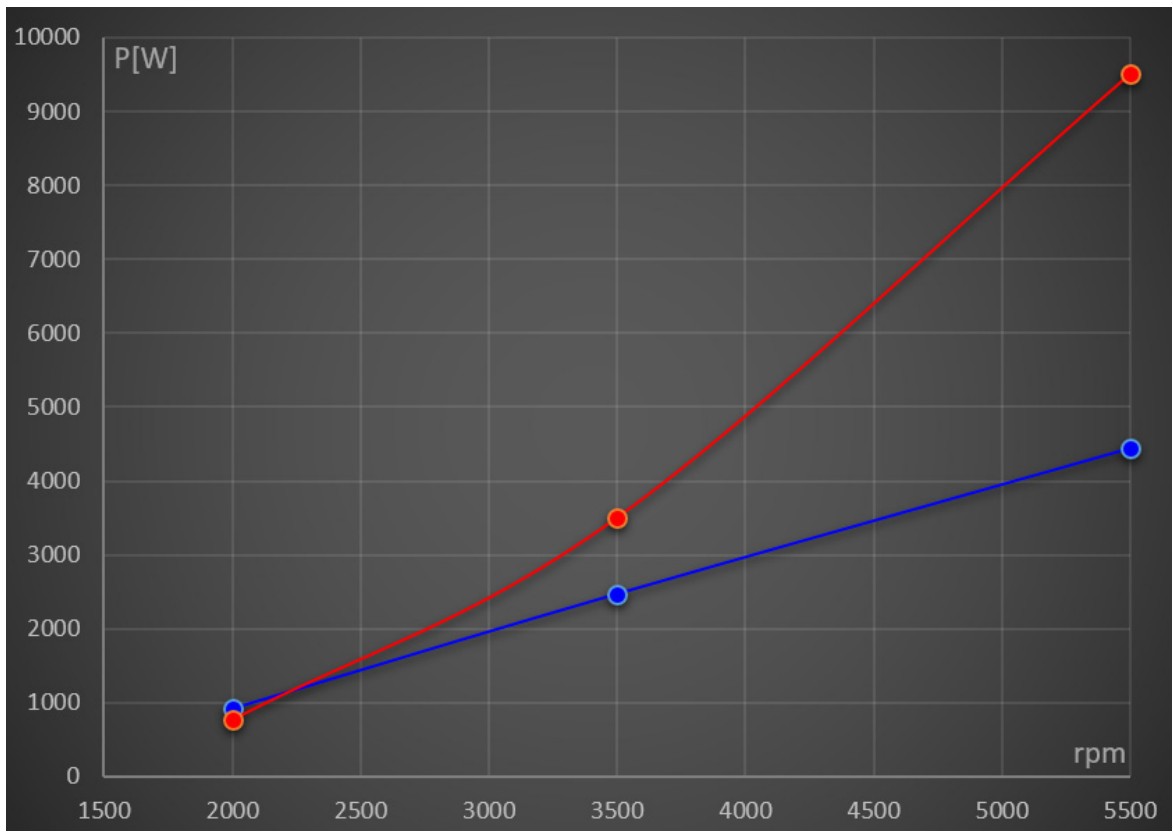

**Figure 10.** Turbine (orange) and compressor (blue) power in function of ICE RPMs.

## 6. Conclusions

The research aimed to test the feasibility of a prototype of a newly designed thermal engine for a hybrid propulsion vehicle. The preliminary proposal, here reported, has been evaluated. The feasibility of the system has been demonstrated. The concept to mechanically disconnect the compressor/turbine device, supporting the rotation of the compressor with a dedicated electric motor and connecting a turbine to a generator has been proved. Besides, the turbomachinery components (compressor and turbine) have been redesigned ex novo, because the existing group on the vehicle corresponds to a GT12 turbocharger and is designed for that purpose. Decoupling the system, to perform—turbine side—a surplus of power for being able to meet the compressor request, and at the same time providing extra power to the vehicle (to the vehicle system) it was necessary to re-design the turbine and make some slight changes. Fortunately for the turbine, it has seen that it corresponds to a GT20 and this allows one to use parts already produced with undoubted economic savings. Similar considerations exist for the compressor. A possible problem can be derived from the available space to insert, within the vehicle, the compressor and its inducer modification and coolant circuit for bearings (due to rpm), electric motor and turbine and its nozzle modification, coolant circuit for bearings and the electric generator. Standard design methods have been used. Further finite elements (FEM) and CFD simulations will lead to system improvements and confirm the goodness of the proposed choice. This new configuration will decrease fuel consumption, increasing emissions abatement, and improving overall vehicle performance. This presented solution can be considered as a mid-term solution for the transportation system, toward pure full

electric vehicles. The final step of this research will be the construction and assembly of the "new" turbocharger device and tested it on the ICE, in the University Laboratory.

**Funding:** This research received no external funding.

**Conflicts of Interest:** The authors declare no conflict of interest.

## Abbreviations

| | |
|---|---|
| c | Specific Heat [J/kg K] |
| CFD | Computational Fluid Dynamic |
| FEM | Finite Elements |
| GT | Gas Turbine |
| ICE | Internal Combustion Engine |
| KERS | Kinetic Energy Recovery System |
| m | Mass Flow rate [kg/s] |
| N | Number of stages |
| n | Rotational speed |
| p | Pressure [Pa] |
| P | Power [W] |
| Q | Volumetric Flow rate [m$^3$/s] |
| r | Radius [m] |
| R | Reaction degree |
| rpm | Revolutions per minute |
| U | Peripheral Velocity [m/s] |
| V | Absolute Velocity [m/s] |
| T | Temperature [K] |
| TC | Turbo compressor |
| W | Work [J/kg] |
| **Greek Letter** | |
| β | Compression ratio |
| χ | Hub to shroud ratio |
| δ | Blockage factor |
| η | Efficiency |
| φ | Flow Coefficient |
| ρ | Density [kg/m$^3$] |
| ψ | Load Coefficient |
| ω | angular speed [rad/s] |
| **Subscripts** | |
| 1 | Inlet/Outlet section |
| 2 | Inlet/Outlet section |
| EUL | Eulerian |
| in | Inlet |
| out | Outlet |
| s | Specific |

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
