# Peer review of "Preliminary Analysis of a New Power Train Concept for a City Hybrid Vehicle"

_designs, 2021_

Round 1

Reviewer 1 Report

Following suggestions are made to improve the quality of a good paper.

  1. Both the abstract and conclusion should include quantitative details of the prototype of the newly designed thermal engine for hybrid propulsion vehicle especially in terms of proposed system's feasibility gauged by parameters pertaining to the performance of compressor, turbine and overall engine operation.
  2. The literature review should highlight the novelty/significance of the proposed prototype of a newly designed thermal engine in light of the research gap in the similar investigations, such as:
    • R. Beck et al., "Model Predictive Control of a Parallel Hybrid Vehicle Drivetrain," Proceedings of the 44th IEEE Conference on Decision and Control, Seville, Spain, 2005, pp. 2670-2675, doi: 10.1109/CDC.2005.1582566.
    • Leontopoulos C, Etemad MR, Pullen KR, Lamperth MU. Hybrid vehicle simulation for a turbogenerator-based power-train. Proceedings of the Institution of Mechanical Engineers, Part D: Journal of Automobile Engineering. 1998;212(5):357-368. doi:10.1243/0954407981526028

Author Response

About the quantitative values, all depending on the results of experimental campaign, that will start in April 2021. The only data supplied derived from the design procedure and are and remain “previewed values”.

As far as literature and its review are concerned, the two references given are very marginal, as one describes the simulation model of a parallel hybrid vehicle. In this paper the battery model (i.e.) is a model that is no longer used. The second paper describes a power train with two electric motors a gearbox for the parallel that is not comparable with the solution proposed here. They have also been included in the bibliography but not mentioned

Reviewer 2 Report

line:

15: How do you quantify "slightly"?

46: refraze "situated immediately".

93:"You mean "engine" capacity not "vehicle".

132: Do the 5 variables be in a given range? If "yes", please specify.

182: According to line 35, the compressor kicks in at 60k-100k rpm. Then why the calculated "n" is 140k-200k rpm?

206: Table 1, please re-arrange for better understanding. Similar to table 2.

280: same as 132

283: What means "temptative"? 

345: same as 206 regarding table 3

434: Why to re-design components when there are similar ones already existing?

General observation: Regarding both turbo-compressor and turbine, the author has set the values for the variables that the end result is similar to products already existing on the market (lines 203-204 for compressor and 347 for turbine), as: Garrett GT12 for turbocompressor and Garrett GT20 for turbine. So, why the trouble?

In my opinion, the activity to calculate new compressor and turbine and obtaining in fact similar products to already existing ones is redundant thus having no novelty. 

Author Response

15: How do you quantify "slightly"?

On the existing turbocharger the turbine rotational speed varies from 60-190K rpm, in our case the turbine speed is about 160K, about 15% lower

46: refraze "situated immediately".

Done, thanks

93:"You mean "engine" capacity not "vehicle".

Figure out, thanks

132: Do the 5 variables be in a given range? If "yes", please specify.

Yes, the chosen values are described in the text, thanks

182: According to line 35, the compressor kicks in at 60k-100k rpm. Then why the calculated "n" is 140k-200k rpm?

The compressor reaches those speeds when connected to the turbine and ICE. The proposal is to decouple it from the turbine and bring it to maximum efficiency for the imposed beta, corresponding to the rotational regimes of the ICE.

206: Table 1, please re-arrange for better understanding. Similar to table 2.

corrected, thanks

280: same as 132

Same answer 132

283: What means "temptative"?

These values are adopted and derived from experience and comparison with existing models. It starts the design procedure - iterative - and sometimes it is need to change or assign a new value to the chosen parameters. They are preliminary, first-attempt, and fixed

345: same as 206 regarding table 3

Done, thanks

434: Why to re-design components when there are similar ones already existing?

Because the existing group on the vehicle corresponds to a GT12 turbocharger and is designed for that purpose. Decoupling the system, in order to have - turbine side - a surplus of power in order to be able to meet the compressor request at the same time and provide extra power to the vehicle (to the vehicle system) it was necessary to resize the turbine and make some slight changes. Fortunately for the turbine we have seen that it corresponds to a GT20 and this allows us to use parts already produced with undoubted economic savings. Similar discourse for the compressor.

General observation: Regarding both turbo-compressor and turbine, the author has set the values for the variables that the end result is similar to products already existing on the market (lines 203-204 for compressor and 347 for turbine), as: Garrett GT12 for turbocompressor and Garrett GT20 for turbine. So, why the trouble?

The possible problem can be derived from the available space to insert, within the vehicle, the compressor and its inducer modification + coolant circuit for bearings (due to rpm) +electric motor and turbine and its nozzle modification + coolant circuit for bearings + electric generator

Round 2

Reviewer 2 Report

There are no reference citations within: 1. Introduction and 2. The proposed task, even though the current state-of-the-art is described. Please address this issue.

284: please try to replace the word: "temptive" with the provided answer:

"These values are adopted and derived from experience and comparison with existing models. It starts the design procedure - iterative - and sometimes it is need to change or assign a new value to the chosen parameters. They are preliminary, first-attempt, and fixed"

410: please insert the answers: 

"Because the existing group on the vehicle corresponds to a GT12 turbocharger and is designed for that purpose. Decoupling the system, in order to have - turbine side - a surplus of power in order to be able to meet the compressor request at the same time and provide extra power to the vehicle (to the vehicle system) it was necessary to resize the turbine and make some slight changes. Fortunately for the turbine we have seen that it corresponds to a GT20 and this allows us to use parts already produced with undoubted economic savings. Similar discourse for the compressor" and 

"The possible problem can be derived from the available space to insert, within the vehicle, the compressor and its inducer modification+coolant circuit for bearings (due to rpm)+electric motor and turbine and its nozzle modification+coolant circuit for bearings+electric generator" within the text in order to emphasise your work.

Author Response

R: There are no reference citations within: 1. Introduction and 2. The proposed task, even though the current state-of-the-art is described. Please address this issue.

A: Corrected, thanks

R: 284: please try to replace the word: "temptive" with the provided answer:

"These values are adopted and derived from experience and comparison with existing models. It starts the design procedure - iterative - and  They are preliminary, first-attempt, and fixed"

A: Done, Thanks

R: 410: please insert the answers: 

"Because the existing group on the vehicle corresponds to a GT12 turbocharger and is designed for that purpose. Decoupling the system, in order to have - turbine side - a surplus of power in order to be able to meet the compressor request at the same time and provide extra power to the vehicle (to the vehicle system) it was necessary to resize the turbine and make some slight changes. Fortunately for the turbine we have seen that it corresponds to a GT20 and this allows us to use parts already produced with undoubted economic savings. Similar discourse for the compressor" and 

"The possible problem can be derived from the available space to insert, within the vehicle, the compressor and its inducer modification+coolant circuit for bearings (due to rpm)+electric motor and turbine and its nozzle modification+coolant circuit for bearings+electric generator" within the text in order to emphasise your work.

A: Done, Thanks
